# A Novel Approach to Reducing Lung Metastasis in Osteosarcoma: Increasing Cell Stiffness with Carbenoxolone

**Kouji Kita [1] [iD], Kunihiro Asanuma [1],* [iD], Takayuki Okamoto [2] [iD], Eiji Kawamoto [3] [iD], Koichi Nakamura [1], Tomohito Hagi [1], Tomoki Nakamura [1] [iD], Motomu Shimaoka [3] and Akihiro Sudo [1] [iD]**

1   Department of Orthopedic Surgery, Mie University Graduate School of Medicine, 2-174 Edobashi, Tsu 514-8507, Mie, Japan; kkita125@gmail.com (K.K.); a-sudou@clin.medic.mie-u.ac.jp (A.S.)
2   Department of Pharmacology, Faculty of Medicine, Shimane University, 89-1 Enya-cho, Izumo-shi 693-8501, Shimane, Japan
3   Department of Molecular Pathobiology and Cell Adhesion Biology, Mie University Graduate School of Medicine, 2-174 Edobashi, Tsu 514-8507, Mie, Japan; shimaoka@doc.medic.mie-u.ac.jp (M.S.)
*   Correspondence: kasanum@gmail.com; Tel.: +81-59-231-5022; Fax: +81-59-231-5211

**Abstract:** Aim: Primary malignant bone tumor osteosarcoma can metastasize to the lung. Diminishing lung metastasis would positively affect the prognosis of patients. Our previous studies demonstrated that highly metastatic osteosarcoma cell lines are significantly softer than low-metastasis cell lines. We therefore hypothesized that increasing cell stiffness would suppress metastasis by reducing cell motility. In this study, we tested whether carbenoxolone (CBX) increases the stiffness of LM8 osteosarcoma cells and prevents lung metastasis in vivo. Methods: We evaluated the actin cytoskeletal structure and polymerization of CBX-treated LM8 cells using actin staining. Cell stiffness was measured using atomic force microscopy. Metastasis-related cell functions were analyzed using cell proliferation, wound healing, invasion, and cell adhesion assays. Furthermore, lung metastasis was examined in LM8-bearing mice administered with CBX. Results: Treatment with CBX significantly increased actin staining intensity and stiffness of LM8 cells compared with vehicle-treated LM8 cells ($p < 0.01$). In Young's modulus images, compared with the control group, rigid fibrillate structures were observed in the CBX treatment group. CBX suppressed cell migration, invasion, and adhesion but not cell proliferation. The number of LM8 lung metastases were significantly reduced in the CBX administration group compared with the control group ($p < 0.01$). Conclusion: In this study, we demonstrated that CBX increases tumor cell stiffness and significantly reduces lung metastasis. Our study is the first to provide evidence that reducing cell motility by increasing cell stiffness might be effective as a novel anti-metastasis approach in vivo.

**Keywords:** metastasis; cell stiffness; atomic force microscopy; carbenoxolone; actin cytoskeleton





## 1. Introduction

Osteosarcoma is a common type of bone cancer. It is the most frequently occurring primary malignant tumor in bones. Despite advances in treatment, half of all patients with osteosarcoma develop lung metastasis, which is a major cause of death. [1]. Metastasis is responsible for over 90% of cancer-related deaths, including deaths due to not only osteosarcoma but a wide variety of tumors. Therefore, preventing the metastasis of tumor cells would have beneficial effects for patients diagnosed with cancer [2]. Especially in regard to osteosarcoma patients, the ability to predict and prevent lung metastasis would improve prognosis. Researchers have explored various approaches to prevent osteosarcoma metastasis, such as targeting specific molecules [3]. However, the prognosis for patients with metastatic osteosarcoma has not improved in 30 years [4]. Metastasis involves multiple steps, including release of cancer cells from the primary tumor, destruction of the surrounding tissue, invasion of blood vessels or lymphatics, migration to target organs, and ectopic growth. Among these metastatic cascades, inhibition of dominant molecules

related to the invasion, migration, adhesion, and proliferation of tumor cells resulted in marked reductions in lung metastasis in various in vivo experimental models [5].

In addition to molecular pathways, recent studies have reported that cell deformability affects invasion, migration, adhesion, and proliferation, suggesting that the stiffness of cancer cells plays a role in metastasis. Cell stiffness, defined as Young's modulus, is a physical property of cells. Holenstein et al. reported that highly metastatic osteosarcoma cells are softer than their low-metastatic counterparts in studies using two paired cell lines, a low-metastatic parental osteosarcoma cell line (SaOs-2) and an experimentally derived highly metastatic variant line (LM5). The metastatic phenotype of LM5 cells is experimental lung metastasis [6]. Collectively, available data suggest that highly metastatic tumor cells are comparatively softer than non-malignant cells [7]. However, it remains unclear whether artificial regulation of malignant cell stiffness could prevent metastasis in vivo.

We previously investigated tumor cell stiffness by comparing highly metastatic LM8 mouse osteosarcoma cells to low-metastatic Dunn cells. Dunn cells exhibit spontaneous lung metastasis, whereas LM8 cells derived from Dunn cells exhibit a metastatic phenotype acquired through repeated in vivo selection. Our data demonstrated that the stiffness of LM8 cells is significantly lower than that of Dunn cells and that the degree of cell stiffness is correlated with cell motility and metastasis [8]. Additionally, our data showed that F-actin is related to cell stiffness [8]. These results suggest that agents that increase cell stiffness might be useful for the prevention of osteosarcoma lung metastasis.

Recently, Okamoto et al. reported that gap junction (GJ)-mediated cell–cell interactions play an important role in the regulation of cell stiffness [9]. GJ channels are formed by members of the connexin (Cx) family, and these channels are essential for the intercellular transport of small molecules among neighboring cells. Studies have demonstrated that inhibition of GJs by chemical reagents or anti-Cx antibodies increases endothelial cell stiffness [9]. The GJ inhibitor carbenoxolone (CBX) [10], derived from glycyrrhizine, is prescribed for treating inflammation and esophageal ulcers [11]. As CBX treatment has been shown to increase cell stiffness, we examined whether CBX treatment can lead to suppression of metastasis related to cell stiffness. We investigated the effect of CBX on tumor cell stiffness, proliferation, invasion, and adhesion. Furthermore, we evaluated whether the administration of CBX reduces lung metastasis using in vivo mouse models.

## 2. Materials and Methods

### 2.1. Cell Culture

The highly metastatic LM8 osteosarcoma cell line was derived from the Dunn osteosarcoma cell line through eight repeated cycles of the procedure described by Poste and Fidler [12]. LM8 mouse osteosarcoma cells (Suita, Osaka, Japan, December 2014) were kindly provided by Osaka University [13]. The Dunn cells were maintained in Minimum Essential Medium (MEM: Gibco BRL, Grand Island, NY, USA), while the LM8 cells were maintained in Dulbecco's modified Eagle's medium (Gibco BRL, Grand Island, NY, USA) supplemented with 10% fetal bovine serum. Cells were cultured at 37 °C in an incubator under 5% $CO_2$.

### 2.2. Fluorescence Staining and Analysis of F-Actin

The actin cytoskeleton was evaluated as previously described [8]. Cells were grown on a glass coverslip to a density of $3.0 \times 10^4$ cells/mL with 500 µL of cell culture medium. After 48 h, the cells were rinsed with phosphate-buffered saline (PBS), and CBX was added at a concentration of 2 µM in cell culture medium. Cells were incubated at 37 °C for 24 h, and the actin cytoskeleton was stained with Alexa Fluor 488-conjugated phalloidin (Thermo, Tokyo, Japan, Molecular Probes). The cells were then fixed with 4% paraformaldehyde containing 0.1% Triton-X for 30 min and blocked with 1% bovine serum albumin (BSA) for 30 min at room temperature and subsequently incubated for 20 min with Alexa Fluor 488-conjugated phalloidin in 1% BSA. All images were captured using a microscope (Olympus, Tokyo, Japan) at an excitation wavelength of 488 nm with a 100 ms exposure time. Schaub

reported that the fluorescence intensity of phalloidin-stained images is proportional to the density of actin filaments and that variations in staining intensity reflect variations in actin density [8]. We used ImageJ software (version 1.52a) (NIH Image, Bethesda, MD, USA) to measure the mean intensity of F-actin staining [8]. To assess intensity variations, we used the ratio of standard deviation (SD) of intensity to the mean intensity of the image. Contrast is expected to be proportional to the inverse of the square root of the concentration, which was calculated using a previously reported formula [8].

### 2.3. Measurement of Cell Stiffness

In this study, atomic force microscopy (AFM) was used to measure cell stiffness. A NanoWizard 3 AFM system (JPK Instruments AG, Berlin, Germany) with a cantilever and tetrahedral-type probe (BL-AC40TS-C2; Olympus, Tokyo, Japan) was employed. AFM was used to measure the mechanical properties of cells. Cell stiffness was defined according to Young's modulus. Measurement of cell stiffness using AFM was performed as previously described [13]. Young's modulus was used as an index of cell stiffness [14]. The cells were grown on glass-bottomed culture dishes to a density of $3.0 \times 10^4$ cells/mL and incubated at 37 °C for 48 h. After rinsing with PBS, CBX was administered at a concentration of 2 μM. Thereafter, the cells were incubated at 37 °C for 24 h, and measurements were conducted in cell culture medium at room temperature. All force curves and scanning field images (10 μm × 10 μm) were recorded at a resolution of 128 × 128 pixels in Quantitative Imaging (QI) mode at 37 °C. The maximum force and contact point determined by the vertical deflection of the cantilever was set to 0.5 nN, and the scan rates were automatically controlled by the Z length (1 μm), extension time (15 ms) and retraction time (15 ms). The data were processed via curve-fitting with the Hertz contact model using JPK data processing software. The geometric mean of Young's modulus was calculated from the acquired Young's modulus at each point of the cell under each given condition [8].

### Evaluation of Cell Processes In Vitro

We evaluated metastasis potential using the LM8 highly metastatic cell line. In this study, we evaluated whether CBX was involved in the proliferation, migration, invasion, and adhesion potential. CBX was administered at a concentration of 2 μM.

### 2.4. Proliferation Assay

Cell proliferation potential was determined using an MTS assay. Untreated control and 2 μM CBX-treated LM8 cells were seeded at a volume of 100 μL and density of $3.0 \times 10^4$ cells/mL and incubated at 37 °C. After 24, 48, and 72 h, cell proliferation was measured using the CellTiter 96™ AQueous non-radioactive cell proliferation assay (Promega, Mannheim, Germany).

### 2.5. Migration Assay

Cell migration potential was assessed using a wound-healing assay. LM8 cells were seeded and cultured to reach confluency, and then the layer of cells was scratched with a 10 μL pipette tip and imaged. After rinsing with PBS, CBX was administered at a concentration of 2 μM. Images were acquired using a microscope (Olympus, Tokyo, Japan) (×200) at 0, 24, 48, and 72 h post-wounding and analyzed using ImageJ software. The rate of cellular migration was determined as cells moving from the intact zones into the scratched region at the different time points. This evaluation method set the cell-free area at 0 h to 100% and measures the rate of reduction in the cell-free area.

### 2.6. Invasion Assay

Invasion potential was analyzed using Matrigel invasion chambers (BD Pharmingen, Corston, UK) in 24-well plates. LM8 cells were seeded in control and drug administration groups in the upper chamber in a 500 μL volume at a density of $5.0 \times 10^4$ cells/mL. The lower chamber was filled with medium. The cells were incubated for 48 h at 37 °C in

an incubator under 5% $CO_2$. Invading cells migrate in a vertical direction through the pores of the membrane of the upper chamber into the lower chamber, in which medium is present. After rinsing with PBS, the filter was fixed and stained with hematoxylin and eosin. Invasion potential was determined by counting the stained cells on the bottom surface of the upper chamber following imaging of three random fields of invading cells using a microscope ($\times$200).

### 2.7. Adhesion Assay

Cell adhesion potential was analyzed using 96-well V-bottom plates. In this assay, centrifugal force is applied to separate adherent from nonadherent cells. The force produced by the centrifugation step results in the accumulation of free or loosely attached cells in the bottom of the V-shaped wells. Nonadherent cells are quantified using a fluorescence reader. The V-bottom plates were blocked with 1% sterile BSA in PBS (200 µL) for 1 h at 37 °C in an incubator. The cells were then labeled with 2′,7′-bis(2-carboxyethyl)-5,6-carboxyfluorescein (BCECF) at 1 µL/mL for 30 min at 37 °C. These cells were diluted with HBS and 100 µL of BCECF-labeled cells was dispensed in each well to provide a density of $9.0 \times 10^4$ cells/mL. The plate was then centrifuged at $280 \times g$ (1200 rpm) for 5 min using a swinging bucket rotor (EX-125; Tomy Seiko Co., Ltd., Osaka, Japan). Nonadherent cells accumulated in the nadir of the V-bottom wells and were quantified using a 2030 ARVO X-2 multilabel reader (PerkinElmer Japan Co., Ltd., Kanagawa, Japan) [15].

### 2.8. In Vivo Lung Metastasis Mouse Model

C3H/He mice were purchased from JAPAN SLC Inc. (Shizuoka, Japan). Mice were used according to guidelines approved by our institution. The study protocol was approved by the Institutional Ethics Review Board. Mice were randomly divided into three groups (n = 8/group), and the experimental model was established. After suspending LM8 cells in PBS, 200 µL of cell suspension ($1 \times 10^7$ cells) was injected into the dorsal subcutaneous area of C3H/He female mice. Treatment was initiated 1 day after tumor cell injection with daily dorsal subcutaneous injections of CBX at 0.5 mg/kg, 5 mg/kg, or saline (control). The tumor volume and weight of the mice were evaluated twice a week. Tumor volume was calculated using the following formula: length $\times$ width$^2$ $\times$ 0.52 [16]. The mice were sacrificed at 5 weeks after injection, and the lungs and tumors were carefully excised. The excised lungs and tumors were fixed with formalin, embedded in paraffin, sectioned, and stained with hematoxylin and eosin for histological observation. Lung metastasis was assessed by counting the metastatic nodes using hematoxylin and eosin staining (Figure 1).

### 2.9. Statistical Analyses

In vitro and vivo data are presented as the mean $\pm$ SD. Statistical analysis of differences in mean values was conducted using nonparametric analysis with the Wilcoxon–Mann–Whitney test. For all statistical tests, a *p* value < 0.05 was considered significant. Statistical analyses were performed using the EZR graphical user interface [17].

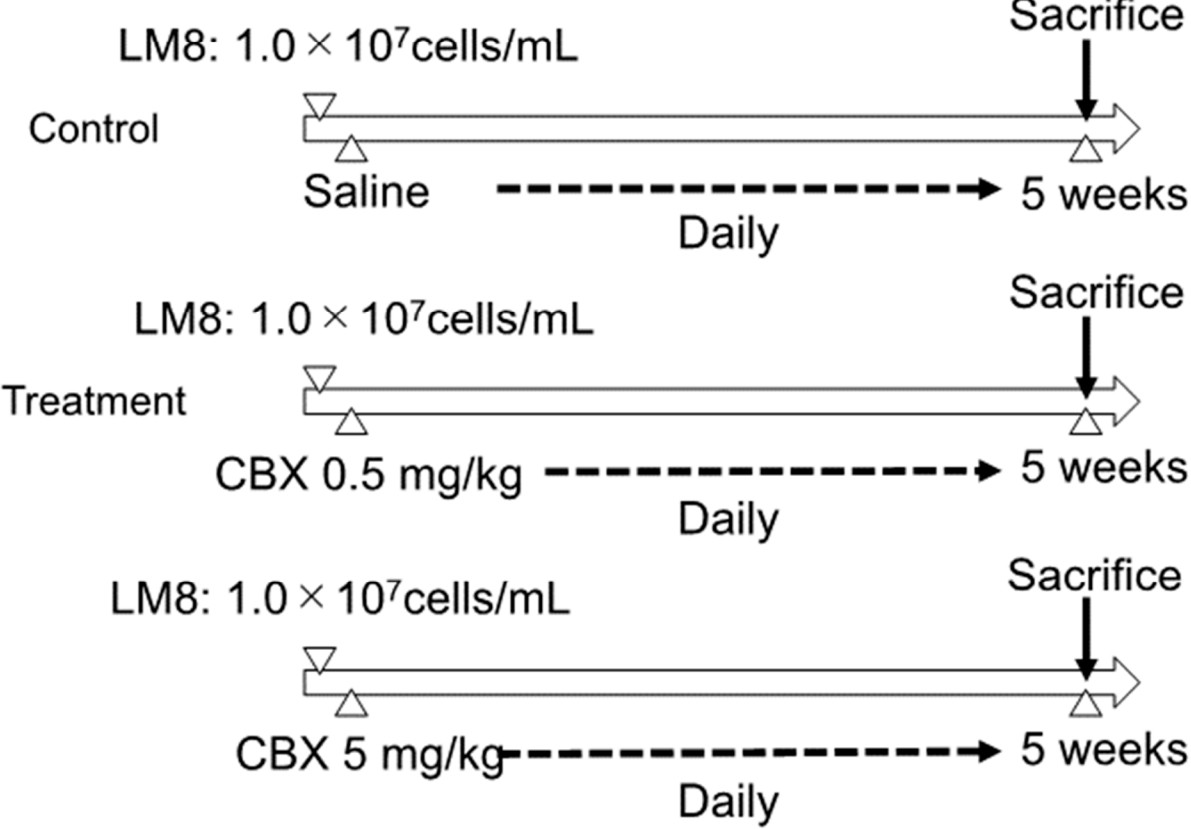

**Figure 1.** Establishment of the experimental model and protocol. Mice were randomly divided into three groups (n = 8/group). LM8 cells were suspended in PBS and then injected into the dorsal subcutaneous area of C3H/He female mice. Daily treatment was initiated 1 day after tumor cell injection. Mice were sacrificed at 5 weeks after injection, and the lungs and tumors were excised. The lung metastasis was measured by counting the metastatic node using hematoxylin and eosin staining.

### 3. Results

*3.1. CBX Increases Cell Stiffness and Actin Polymerization*

We used AFM to measure cell stiffness in both the LM8 (control) and CBX treatment groups, comparing the geometric mean of Young's modulus for each group. We also evaluated stiffness of Dunn as a control for low metastatic cells. Cell stiffness was determined by analyzing the obtained force curves and reconstructing them as stiffness images. The mean stiffness of the control group was $8.20 \pm 0.36$ kPa, and that of the CBX treatment group was $8.88 \pm 0.16$ kPa. The mean stiffness of Dunn, a low metastatic cell, was $8.95 \pm 0.39$ kPa. Cells of the CBX treatment group were significantly stiffer than cells of the control group ($p < 0.01$) (Figure 2a). In Young's modulus images, compared to the control group, more rigid fibrillate structures were observed in the CBX treatment group (Figure 2b,c). We also investigated the change in actin polymerization and distribution of LM8 cells using CBX to examine changes in actin polymerization. In fluorescence images of F-actin, the actin filaments in the control group were shorter, less organized, and oriented randomly. Conversely, in comparison with the control cells, the actin filaments in cells of the CBX treatment group were distributed throughout the cell and aligned along the long axis of the cells (Figure 3a). We evaluated the intensity of actin staining using ImageJ software. Actin

staining intensity in the CBX treatment group was significantly higher than that of the control group (Figure 3b). These indicate that actin polymerization is strongly involved in determining cell stiffness. Furthermore, analysis of actin staining intensity produced results similar to those observed for cell stiffness. These results indicated that actin changes could account for the observed differences in cell stiffness, and administration of CBX promotes actin polymerization and increases cell stiffness.

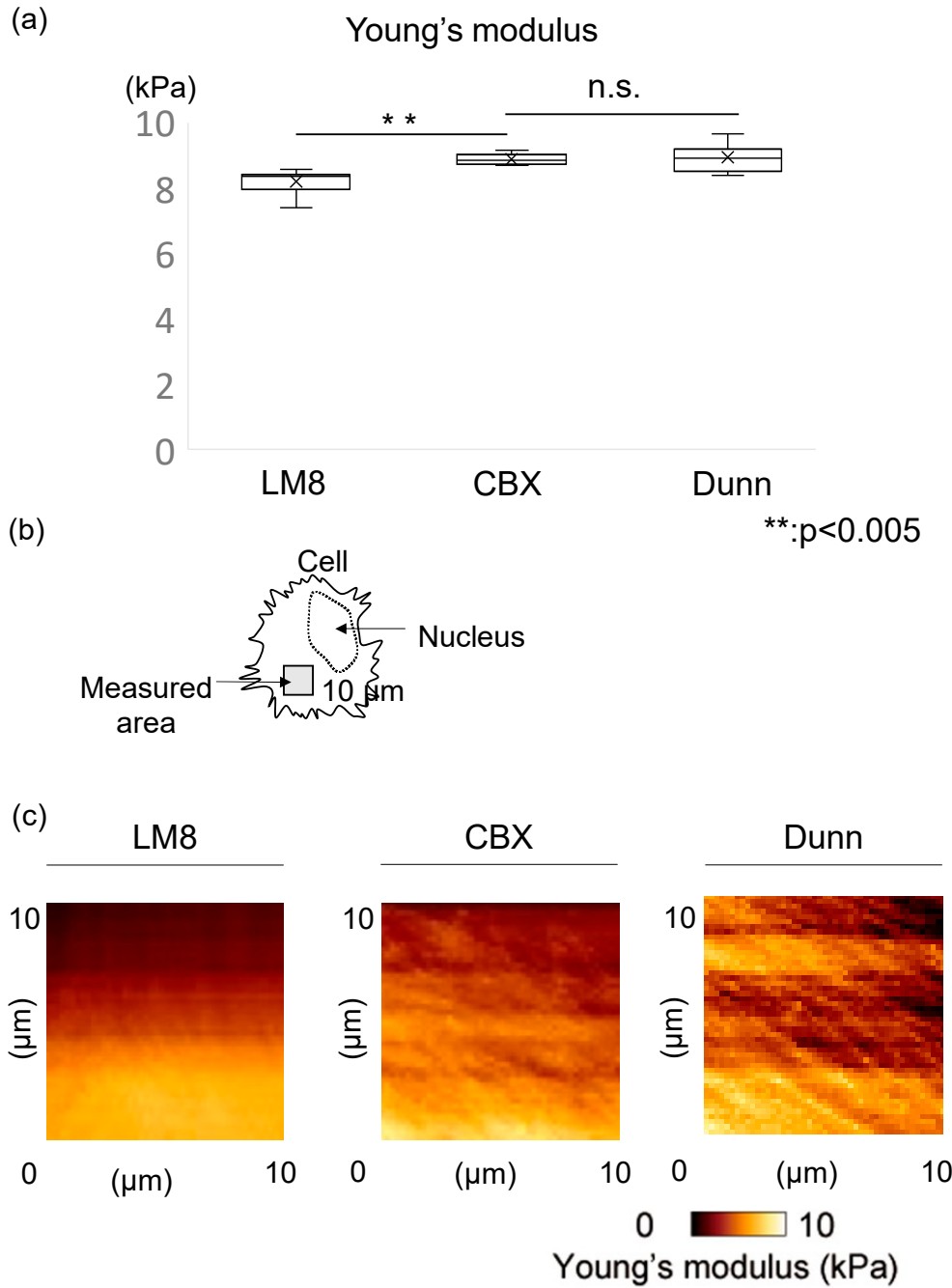

**Figure 2.** Stiffness and actin localization in the cell. (**a**) The Young's modulus was compared between control group (LM8), CBX treatment group, and low metastatic group (Dunn). LM8: $8.20 \pm 0.36$ kPa, CBX: $8.88 \pm 0.16$ kPa, Dunn: $8.95 \pm 0.39$ kPa. CBX treatment group was significantly stiffer than LM8 ($p < 0.01$). In the CBX treatment group, the stiffness was similar to Dunn. (**b**) Measured area using AFM was indicated in the scheme. (**c**) Young's modulus images in control and CBX treatment were shown. In the CBX treatment group, the actin filaments in cells were distributed throughout the cell.

## Fluorescence images of F-actin

(a)

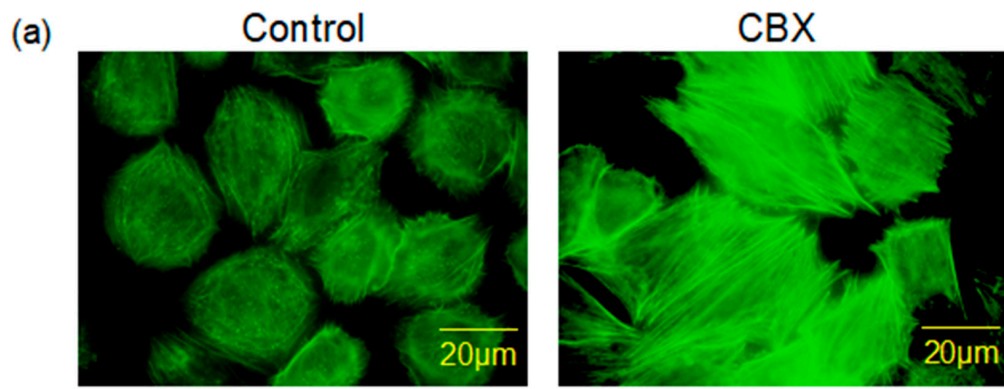

(b)

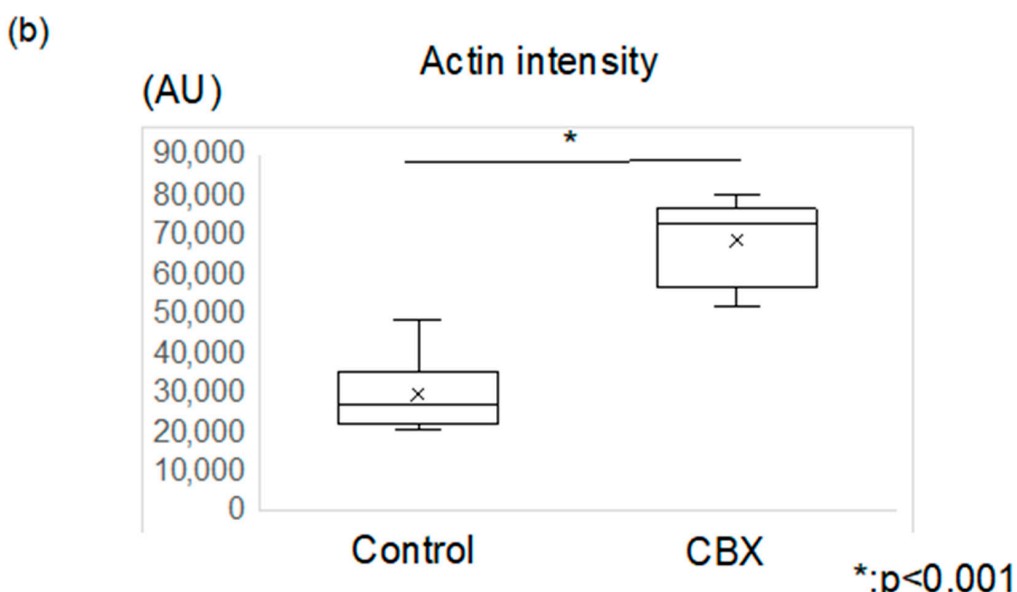

**Figure 3.** Actin polymerization and intensity between control and CBX treatment. (**a**) Actin cytoskeleton was stained with Alexa 488-conjugated phalloidin. The actin filaments in the control group were shorter, less organized, and oriented randomly. Conversely, in comparison with the control cells, the actin filaments in cells of the CBX treatment group were distributed throughout the cell and aligned along the long axis of the cells. (**b**) Actin staining intensity was evaluated via arbitrary units (AU) using Image J software. Actin intensity in the CBX treatment group was significantly higher than that of the control group.

### 3.2. CBX Affects Metastatic Potential

Next, to investigate the impact of CBX on metastatic potential, we evaluated the proliferation, migration, invasion, and adhesion of CBX-treated LM8 cells. Administration of CBX affected not only cell stiffness but also metastatic potential. After 24, 48, and 72 h, cell proliferation and migration potential were measured. In terms of proliferation potential, there was no statistically significant difference between the groups. The migration potential was higher in the control group after 24 h ($p < 0.05$). Significantly lower invasion ($p < 0.005$) and adhesion ($p < 0.05$) potential were observed in the CBX treatment group compared

with the control group (Figure 4). These results suggest that CBX affects the migration, invasion, and adhesion potential of LM8 cells in vitro by altering actin organization.

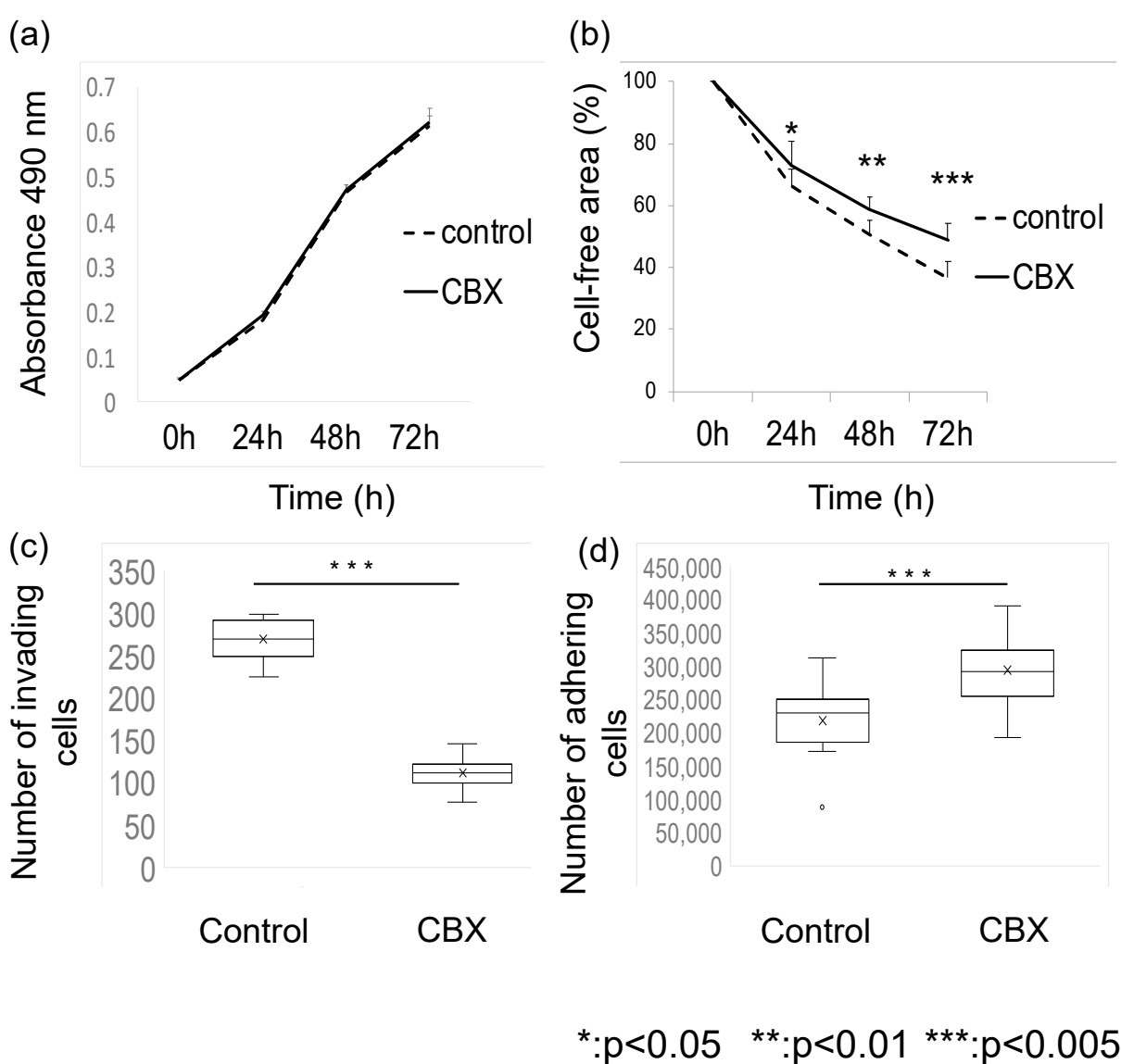

*:p<0.05   **:p<0.01 ***:p<0.005

**Figure 4.** Functional analysis. (**a**) Cell proliferation and (**b**) migration were measured after 3, 24, 48, and 72 h in culture. (**c**) Invasion activity was calculated after 48 h treatment. (**d**) Adhesion activity was evaluated after 1 h treatment. CBX administration did not affect proliferation potential.

### 3.3. CBX Reduces the Number of Lesions in the Lung

We also evaluated whether CBX administration suppresses LM8 cell metastasis to lung tissue in mice. The number of lung metastasis nodules, tumor volume, and body weight was compared between the control group and CBX treatment group. In terms of tumor volume and body weight, there were no statistically significant differences between the groups (Figure 5). We assessed lung metastasis by counting the metastatic nodules. In hematoxylin and eosin staining, many metastatic nodules were found in the control group, whereas only a few nodules were found in the CBX treatment group. The number of lesions in the lung was significantly reduced in the CBX treatment group compared with the control group ($p < 0.01$), but there were no differences in tumor volume as a result of CBX treatment. The number of lesions in the lung decreased as cell stiffness increased (Figure 6).

## (a) Tumor volume

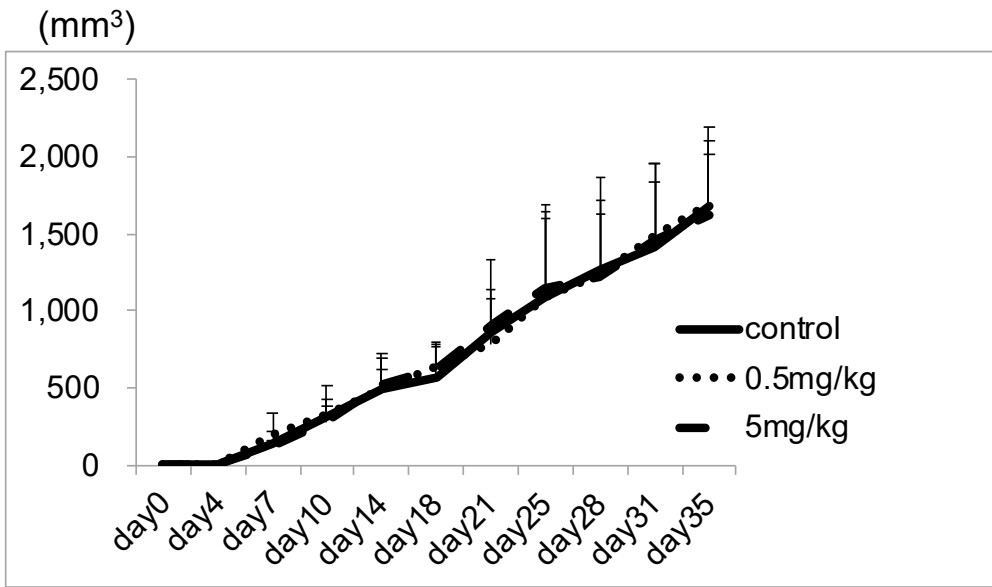

## (b) Body weight

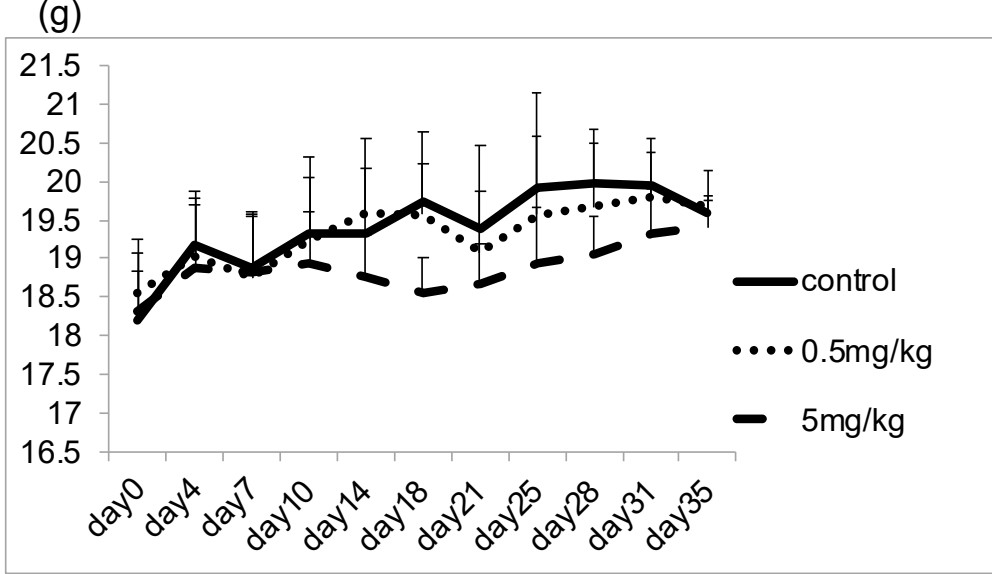

**Figure 5.** Tumor volume and body weight in the control and CBX treatment groups. (**a**) Tumor volume and (**b**) body weight of mice was evaluated twice a week. Tumor volume was calculated using the following formula: length $\times$ width$^2$ $\times$ 0.52. In terms of tumor volume and body weight, there were no statistically significant differences between groups.

## (a) Lung metastasis nodules

Control          CBX: 0.5 mg/kg          CBX: 5 mg/kg

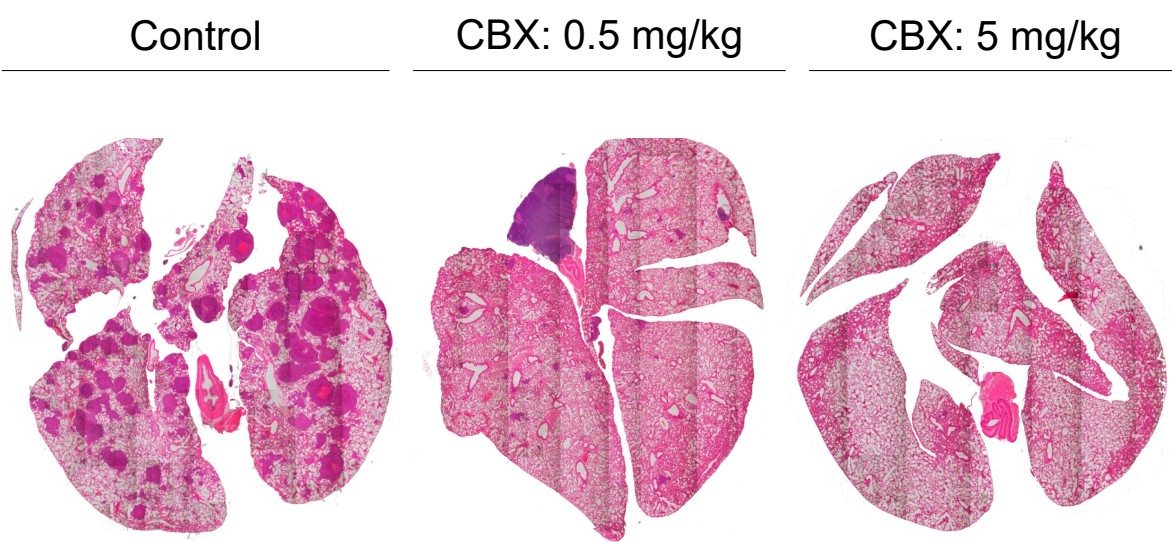

## (b) Number of lesions in the lung

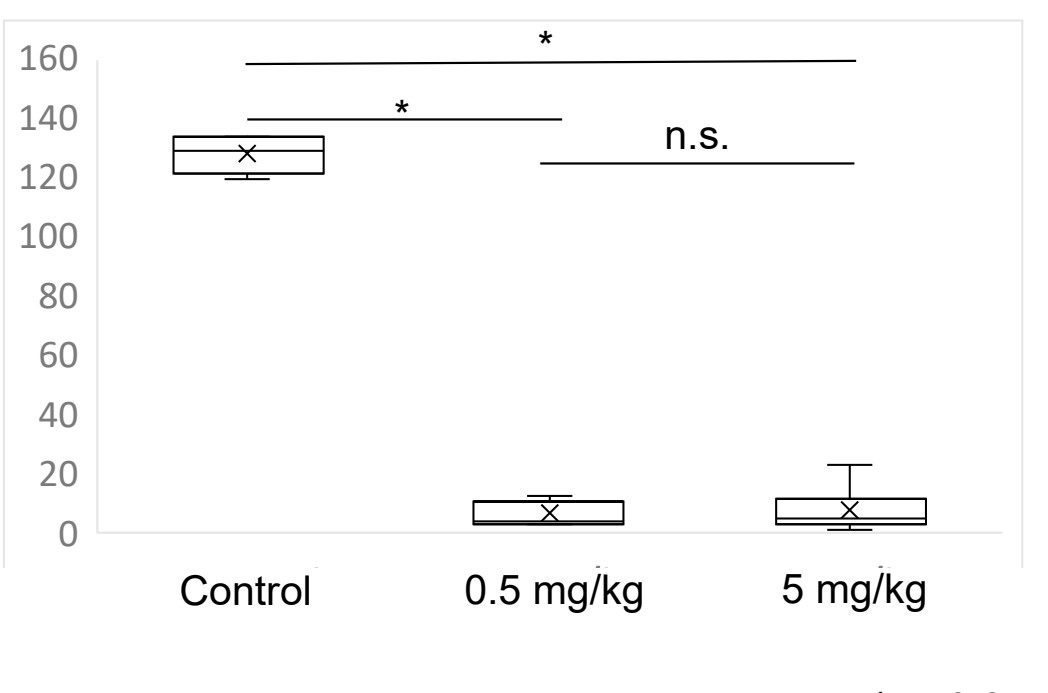

*:p<0.05

**Figure 6.** Evaluation of lung metastases between the control and CBX treatment groups. (**a**) Hematoxylin and eosin staining. Metastatic nodules were counted. (**b**) The number of lesions in the lungs of the control group was significantly higher than that of the CBX treatment group.

## 4. Discussion

Osteosarcoma is the most common primary malignant bone tumor in child and adolescence. Osteosarcoma exhibits highly aggressive and early systemic metastasis. With notable

developments in effective chemotherapy combinations, the survival rates of nonmetastatic osteosarcoma have significantly increased from <20% before 1970s to present rates of 65% to 75%. However, the occurrence of lung metastasis is still the most prominent reason for osteosarcoma-caused death. Only 11% to 30% of patients suffering from osteosarcoma metastasis can survive after the combination of surgery resection and chemotherapy, and thus osteosarcoma metastasis has become the obstacle for successful osteosarcoma treatments [18]. Therefore, in order to profoundly improve clinical outcomes for patients with osteosarcoma, it is not only necessary to find out the metastasis mechanism but also to identify factors that can be used as therapeutic targets in osteosarcoma. Metastasis is a common and critical phenomenon that has been studied extensively; however, the underlying mechanisms involved remain to be fully elucidated and a large scale of regions in molecular mechanisms involved in osteosarcoma metastasis are still uncovered. In the past, motility, migration, invasion, and proliferation have been used to evaluate metastatic potential in vitro. In recent years, attention has been focused on the relationship between cell deformability and metastatic potential as a new barometer of metastatic potential. In this study, we focused on the stiffness of cells and searched for anti-metastatic medicines.

Measuring cell stiffness using AFM has the advantage of being able to quantitatively determine cell stiffness of individual living cells in physiological environments. However, the disadvantage is that measurements require a long time and can only be performed on a limited number of cells. Measurement of actin staining intensity is easier and less time-consuming than measuring Young's modulus using AFM, and can be used as a first step in identifying drug candidates that potentially alter Young's modulus. In this study, we successfully showed a correlation between actin levels and cell stiffness in both the control and CBX administration group.

The occurrence of lung metastasis has a poor prognosis, thus the ability to delay or prevent metastasis has the potential to enormously improve the survival durations of patients with cancer and could even lead to cures. For that reason, the development of new effective medicines to improve prognosis that interrupt the primary causes of metastasis is a major challenge. Various treatments for cancer have been evaluated, such as chemotherapies, hormone therapies, molecular-targeted agents, immunotherapies, and various combination treatments, but there is no standard treatment because metastatic tumor cells are genetically unstable [19]. Metastasis is a multistage process, and various points of intervention have been identified, including the initial steps of invasion and migration away from the primary tumor, entry into the circulation, and extravasation at distant sites.

As with invasion, traversal of the circulatory system might also be an early event in cancer development. Therefore, to conceptualize a new anti-metastatic agent, uncertainties regarding when tumor cells invade the circulation and when metastases form must be considered. Anti-metastasis therapies must be applied immediately when malignant tumors are identified. Thus, we formulated the following concept for the development of anti-metastatic agents: (1) treatment can be started immediately and carries a low risk of side effects; (2) treatment can continue even during surgery or chemotherapy for several months. Based on this framework, we have screened for potential agents from among common agents not used for cancer treatment, such as anti-inflammatory drugs, stomach drugs, and channel blockers.

Existing anti-cancer agents are generally used to reduce the size of target lesions, as determined clinically via image assessment. Reducing tumor lesions uniformly without a distinction between primary, recurrent, or metastatic lesions is critically important. Therefore, anti-cancer agents may prevent metastasis by reducing tumor cell proliferation. However, such an anti-proliferation-related anti-metastatic effect is not direct. Metastasis of cancer cells to distal sites is associated with poor prognosis and is the primary cause of cancer-related death [20]. However, no standard treatments to prevent metastasis are currently available. Additionally, few studies have specifically examined the biological development of metastasis. Clinical drug development generally relies on the demonstra-

tion of tumor shrinkage according to the radiological Response Evaluation Criteria for Solid Tumors criteria, with confirmatory improvements in clinical outcomes, ignoring the ability to inhibit metastasis. Development of an effective anti-metastatic agent requires analysis of mechanisms that affect factors other than cell viability. Therefore, we focused on the stiffness of cells and investigated whether changing cell stiffness affects metastasis. We previously reported that cells with a highly metastatic phenotype exhibit significantly lower stiffness than cells with a low-metastatic phenotype [8]. In this study, we focused on identifying drugs that increase cell stiffness, as stiffer cells may exhibit lower metastatic potential. Screening for drugs that increase cell stiffness is a new approach to drug development that has potential for identifying new therapeutic agents targeting metastasis. Cells treated with CBX were significantly stiffer than control cells, and the number of lesions in the lung decreased as cell stiffness increased. These results demonstrate that controlling cell stiffness can be effective in suppressing metastasis.

The mechanism by which CBX treatment and GJ inhibition control cell stiffness and tumor metastasis is not fully understood. Cell stiffness is affected by contraction forces generated by the actomyosin cytoskeleton and F-actin [21]. The critical roles of Cx and cytoskeletal proteins must be considered when evaluating cell stiffness. Cx can be connected to actin via the zonula occludens 1 (ZO-1)–vinculin complex. ZO-1 belongs to the membrane-associated guanylate kinase (MAGUK) superfamily and has an actin-binding domain within its C-terminus (CT) portion, a vinculin-binding domain at the third PSD-95/Dlg/Zo-1 (PDZ), and an α,β–catenin binding domain at the guanylate kinase (GUK) in the N-terminus (NT) portion [22]. ZO-1 plays a central role in connecting Cx to various cytoskeletal proteins, and double-knockout of ZO-1 and ZO-2 was shown to increase cell stiffness [23]. On the other hand, vinculin is a focal adhesion protein that interacts with the actin cytoskeleton. Vinculin-knockout cells and cells with vinculin but lacking the paxillin- and actin-binding sites exhibit reduced cell stiffness [24]. Furthermore, activation of integrin α5β1 directly affects the opening of Cx43 hemichannels [25]. Integrin–adhesion complexes mediate bi-directional transmembrane signaling via focal adhesions [26]. Treatment with an anti-alpha5 integrin antibody was shown to decrease cell stiffness [27]. However, it is still unclear whether inhibition of Cx inactivates integrin.

We also demonstrated that CBX treatment reduced tumor cell invasion and adhesion. Recent studies suggested that Cx43 plays roles in the regulation of various cellular processes, including migration, proliferation, and shape formation [28]. ZO-1 and vinculin are associated with cell adhesion and invasion [29]. CBX may thus affect tumor cell invasion and adhesion via ZO-1 and vinculin. Another study showed that the number of focal adhesion complexes is increased with CBX treatment [9]. These previous studies support our results and show that regulating GJ channels may affect cell motility and tumor metastasis. Given these data, the enhancement of cell stiffness and suppression of invasion and adhesion associated with CBX treatment could involve inhibition of Cx via ZO-1 or the vinculin interface.

In this study, which used a spontaneous lung metastasis model rather than an experimental lung metastasis model, CBX administration significantly reduced the number of lung metastasis nodules. Our results provide the first evidence that CBX administration suppresses the metastasis of osteosarcoma clinically. In addition, CBX is a derivative of glycyrrhizine, which is already used for the pharmacological treatment of inflammation and esophageal ulcers, suggesting CBX administration would carry relatively low risk. However, the principal side effects of the CBX are sodium retention, hypokalemic alkalosis, suppressed plasma renin, and hypertension. Additionally, pseudohyperaldosteronism has been reported with high amounts of glycyrrhetinic acid. It was thought that carbenoxolone appeared not to have intrinsic mineralocorticoid activity but, rather, it enhanced aldosterone action by displacing it from the nonspecific binding sites [30]. Therefore, caution must be exercised when combining carbenoxolone with other medications as some combinations may disturb the electrolyte balance or cause hypertension. Thus, further investigation is necessary in the future.

We speculate that CBX administration may benefit patients suspected of malignant tumors at the initial visit and that co-administration of CBX and anti-cancer drugs may exhibit an additive effect due to difference in pharmacological action. Furthermore, continuance of CBX administration during chemotherapy until primary tumor resection may lead to improved prognosis and a lower risk of metastasis.

## 5. Conclusions

Our present study demonstrated the inhibitory effect of CBX on lung metastasis. Actin levels and cell stiffness may be used as new parameters to determine metastatic potential and as quantitative indicators in the development of new drugs targeting metastasis. Our results also suggest that CBX administration carries a low risk of side effects. These findings suggest a possible clinical scenario in which CBX could be prescribed as a prophylactic treatment to reduce the metastatic burden immediately after diagnosis of a primary malignant tumor which is at risk of spreading to the lungs.

From our concept, if patients are suspected of malignant tumors at the initial visit, administration of CBX can be started which has a low risk of side effects. CBX did not affect cell proliferation and this allows for CBX administration along with anti-cancer drugs. Continuing CBX administration until at least primary tumor resection even in chemotherapy, may reduce metastasis and improve prognosis. This needs further in-depth study.

**Author Contributions:** Main manuscript, K.K. Conceptualization, K.K. and K.A.; Experiments, K.N., E.K. and K.K. Data analysis, T.H. and T.N.; M.S., T.O. and K.A. critically reviewed the manuscript; A.S. supervised this study. All authors have read and agreed to the published version of the manuscript.

**Funding:** This research received no external funding.

**Institutional Review Board Statement:** The study was approved by the Ethics Committee of Mie University (protocol code 2019-35 and date of approval 6 January 2020).

**Informed Consent Statement:** Not applicable.

**Data Availability Statement:** Not applicable.

**Conflicts of Interest:** The authors declare no conflict of interest.

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
