# Peer review of "A Novel Approach to Reducing Lung Metastasis in Osteosarcoma: Increasing Cell Stiffness with Carbenoxolone"

_cimb, doi:10.3390/cimb45050278_

Round 1

Reviewer 1 Report

Comments and Suggestions for Authors

The manuscript by Kouji et al. presents novel findings that increasing the cell stiffness can suppress metastasis by reducing cell motility. The authors tested this hypothesis by evaluating the actin cytoskeletal structure and polymerization of CBX-treated LM8 cells, measuring cell stiffness, and analyzing metastasis-related cell functions such as cell proliferation, wound healing, invasion, and adhesion. The authors also examined lung metastasis in LM8-bearing mice administered CBX. The results showed that CBX treatment significantly increased the stiffness of LM8 cells, suppressed cell migration, invasion, and adhesion, and reduced the number of lung metastases in LM8-bearing mice. The study provides evidence that increasing cell stiffness might be effective as a novel anti-metastasis approach in vivo.

The manuscript is well-written and well-organized, and the data supports the conclusions. However, there are some areas that require improvement in the manuscript. Firstly, the authors should carefully review the text to eliminate any errors. Additionally, the authors should refine their figures to ensure consistent font format, font size, and color to enhance their appearance and readability.

Regarding line 72-73, the authors should add references to support the statement "Studies have demonstrated that inhibition of GJs by chemical reagents or anti-Cx antibodies increases endothelial cell stiffness." 

The mean stiffness of the control group was 8.20±0.36 kPa, and that of the CBX treatment group was 8.88±0.16 kPa. It is unclear whether the change in stiffness is significant or not. To address this, the stiffness of low-grade cells like Dunn cells should be added as a control group, and the stiffness of Dunn cells and CBX-treated cells should be compared.

In Figure 3a, the scale bar is missing, and the authors should add it to improve the clarity of the figure.

The authors should add references to support the statement that F-actin is related to cell stiffness.

To enhance the in vivo experiment, it would be beneficial if the authors could measure the tumor cell stiffness of two different groups.

Although the CBX did not affect the proliferation of LM8 cells, the body weight of mice decreased in the CBX-treated group. Therefore, the authors might need to investigate the cytoxicity of CBX.

Comments on the Quality of English Language

Moderate editing of English language needed.

Author Response

Assigned Editor: Mr. Bruno Zhang

Journal: CIMB (ISSN 1467-3045)

Manuscript ID: cimb-2373991

Type: Article

Number of Pages: 16

Title: Carbenoxolone prevents lung metastasis of LM8 mouse osteosarcoma cells by regulating cell stiffness

Authors: Kouji Kita , Kunihiro Asanuma * , Takayuki Okamoto , Eiji Kawamoto , Koichi Nakamura , Tomohito Hagi , Tomoki Nakamura , Motomu Shimaoka , Akihiro Sudo

Abstract

Aim: Primary malignant bone tumor osteosarcoma occurs lung metastasis. Diminishing the lung metastasis, would have an enormous effect on the prognosis of patients. Previous our studies have demonstrated that the highly metastatic osteosarcoma cell lines had significantly softer than that of low metastatic cell line. Given this, we hypothesized that increase of cell stiffness might conduct metastasis suppression via reducing cell motility. In this study, we test whether carbenoxolone (CBX) increases LM8 osteosarcoma cell stiffness and prevents lung metastasis in vivo.

Methods: We evaluated actin cytoskeletal structure and polymerization of CBX-treated LM8 and by actin staining. Cell stiffness was measured using Atomic Force Microscopy (AFM). The metastasis related cell functions were analyzed by cell proliferation assay, wound healing assay, invasion assay, and cell adhesion assay. Furthermore, we administrated CBX into LM8-bearing mice and then evaluate the lung metastasis.

Results: Treatment with CBX significantly increased actin intensity and cell stiffness of LM8 compared with vehicle treatment LM8 (P<0.01). In young’s modulus images, compared to the control group, rigid fibrillate structures were observed in the CBX treatment group. CBX also suppressed cell migration, invasion, and adhesion, but not cell proliferation. Moreover, the number of LM8 lung metastasis were significantly reduced in the CBX administration group compared with the control group (P<0.01).

Conclusion: In this study, we demonstrate that CBX increases cell stiffness and significantly reduced lung metastasis. Our study provided first evidence that increase of cell stiffness leading to the reduction of cell motility might be effective as a novel anti-metastatic approach in vivo. 

Responses to the reviewers’ comments
Reviewer #1:

Comment 1:

The manuscript is well-written and well-organized, and the data supports the conclusions. However, there are some areas that require improvement in the manuscript. Firstly, the authors should carefully review the text to eliminate any errors. Additionally, the authors should refine their figures to ensure consistent font format, font size, and color to enhance their appearance and readability.

Response 1:

We fixed the pointed-out parts of this manuscript.

Comment 2:

Regarding line 72-73, the authors should add references to support the statement "Studies have demonstrated that inhibition of GJs by chemical reagents or anti-Cx antibodies increases endothelial cell stiffness." 

Response 2:

We added references to support the statement. The reference No is 9. The paper reports that gap junction-mediated cell-cell interactions play an important role in the regulation of endothelial cellular stiffness.

Comment 3:

The mean stiffness of the control group was 8.20±0.36 kPa, and that of the CBX treatment group was 8.88±0.16 kPa. It is unclear whether the change in stiffness is significant or not. To address this, the stiffness of low-grade cells like Dunn cells should be added as a control group, and the stiffness of Dunn cells and CBX-treated cells should be compared.

Response 3:

Thank you for your comment. Added the stiffness data of Dunn cells. The mean stiffness of Dunn was 8.95±0.39 kPa. In line 176, 178, We added the data and sentence. We have also fixed the figure2 and added the results of Dunn.

Comment 4:

In Figure 3a, the scale bar is missing, and the authors should add it to improve the clarity of the figure.

Response 4:

We added scale bar in Figure 3a.

Comment 5:

The authors should add references to support the statement that F-actin is related to cell stiffness.

Response 5:

We added sentence and reference to support the statement that F-actin is related to cell stiffness.

The reference No is 8.

Comment 6:

To enhance the in vivo experiment, it would be beneficial if the authors could measure the tumor cell stiffness of two different groups.

Response 6:

Thank you for your comment.

This experiment did not measure the stiffness of the tumor cells. We are evaluating stiffness at the single cell level, but we believe that evaluation of the stiffness of the tumor cell is also useful. AFM cannot evaluate formalin-fixed cells, so we would like to make this a research topic in the future.

Comment 7:

Although the CBX did not affect the proliferation of LM8 cells, the body weight of mice decreased in the CBX-treated group. Therefore, the authors might need to investigate the cytoxicity of CBX.

Response 7:

Thank you for your comment. In this study, Suppression of lung metastasis was observed even at a concentration of 0.5mg without weight loss. Citoxicity of CBX was not assessed in this study. There are no reports evaluating cytoxicity of CBX in the past. Further evaluation is required.

Thank you for your comment.

We fixed manuscript again.

Reviewer 2 Report

Comments and Suggestions for Authors

In line 119 you mention proliferation into “Evaluation of metastasis potential in vitro”. I suggest to add another topic or change a topic for a general one “Evaluation of cell processes in vitro”.

In lines 191-192 you mention “These results indicated that actin changes could account for the observed differences in cell stiffness, and administration of CBX promotes actin polymerization and increases cell stiffness.” How do you it is polymerization increase or is depolymerization inhibition?

 In general discussion section I recommend to add references that support what it is argued.

In lines 310-313 you mention “From our concept, if patients are suspected of malignant tumors at the initial visit, administration of CBX can be started with a low risk of side effects. CBX did not affect cell proliferation and this may be able to administer CBX with anti-cancer drugs. By continuance of CBX administration until at least primary tumor resection even in chemotherapy, metastasis may be reduced and prognosis may improve.” As it is in conclusion section, I suggest to change it because this was not evaluated in the work.

Author Response

Assigned Editor: Mr. Bruno Zhang

Journal: CIMB (ISSN 1467-3045)

Manuscript ID: cimb-2373991

Type: Article

Number of Pages: 16

Title: Carbenoxolone prevents lung metastasis of LM8 mouse osteosarcoma cells by regulating cell stiffness

Authors: Kouji Kita , Kunihiro Asanuma * , Takayuki Okamoto , Eiji Kawamoto , Koichi Nakamura , Tomohito Hagi , Tomoki Nakamura , Motomu Shimaoka , Akihiro Sudo

Abstract

Aim: Primary malignant bone tumor osteosarcoma occurs lung metastasis. Diminishing the lung metastasis, would have an enormous effect on the prognosis of patients. Previous our studies have demonstrated that the highly metastatic osteosarcoma cell lines had significantly softer than that of low metastatic cell line. Given this, we hypothesized that increase of cell stiffness might conduct metastasis suppression via reducing cell motility. In this study, we test whether carbenoxolone (CBX) increases LM8 osteosarcoma cell stiffness and prevents lung metastasis in vivo.

Methods: We evaluated actin cytoskeletal structure and polymerization of CBX-treated LM8 and by actin staining. Cell stiffness was measured using Atomic Force Microscopy (AFM). The metastasis related cell functions were analyzed by cell proliferation assay, wound healing assay, invasion assay, and cell adhesion assay. Furthermore, we administrated CBX into LM8-bearing mice and then evaluate the lung metastasis.

Results: Treatment with CBX significantly increased actin intensity and cell stiffness of LM8 compared with vehicle treatment LM8 (P<0.01). In young’s modulus images, compared to the control group, rigid fibrillate structures were observed in the CBX treatment group. CBX also suppressed cell migration, invasion, and adhesion, but not cell proliferation. Moreover, the number of LM8 lung metastasis were significantly reduced in the CBX administration group compared with the control group (P<0.01).

Conclusion: In this study, we demonstrate that CBX increases cell stiffness and significantly reduced lung metastasis. Our study provided first evidence that increase of cell stiffness leading to the reduction of cell motility might be effective as a novel anti-metastatic approach in vivo. 

Responses to the reviewers’ comments
Reviewer #2:

Comment 1:

In line 119 you mention proliferation into “Evaluation of metastasis potential in vitro”. I suggest to add another topic or change a topic for a general one “Evaluation of cell processes in vitro”.

Response 1:

Thank you for your comment.

In line 112, I changed a topic “Evaluation of cell processes in vitro”.

Comment 2:

In lines 191-192 you mention “These results indicated that actin changes could account for the observed differences in cell stiffness, and administration of CBX promotes actin polymerization and increases cell stiffness.” How do you it is polymerization increase or is depolymerization inhibition?

Response 2:

Thank you for your comment.

administration of CBX promotes actin polymerization and increases cell stiffness

→administration of CBX increases actin intensity and cell stiffness. In our previous study, cytochalasin, inhibitor of actin polymerization, decreased actin staining intensity and cell stiffness in both LM8 and Dunn cells (8). We guess CBX may promotes actin polymerization.

Comment 3:

 In general discussion section I recommend to add references that support what it is argued.

Response 3:

Thank you for your comment. In discussion section, we added the references.

The reference No are 18 and 30.

Comment 4:

In lines 310-313 you mention “From our concept, if patients are suspected of malignant tumors at the initial visit, administration of CBX can be started with a low risk of side effects. CBX did not affect cell proliferation and this may be able to administer CBX with anti-cancer drugs. By continuance of CBX administration until at least primary tumor resection even in chemotherapy, metastasis may be reduced and prognosis may improve.” As it is in conclusion section, I suggest to change it because this was not evaluated in the work.

Response 4:

Thank you for your comment. In line 323-, We added a sentence.

Thank you for your comment.

We fixed manuscript again.

Reviewer 3 Report

Comments and Suggestions for Authors

The manuscript entitled "Carbenoxolone prevents lung metastasis of LM8 mouse osteosarcoma cells by regulating cell stiffness" explores the relationship between cancer cell stiffness and metastasis, as well as the potential of carbenoxolone (CBX) in preventing lung metastasis in osteosarcoma. The study is motivated by the fact that reducing metastasis can positively impact the prognosis of patients with malignant bone tumor osteosarcoma. To test this hypothesis, the authors evaluate the cell stiffness of LM8 osteosarcoma cells treated with CBX and assess the effects of CBX on metastasis-related cell functions, as well as on lung metastasis in LM8-bearing mice. The results of the study indicate that CBX significantly increases the stiffness of LM8 cells, suppresses cell migration, invasion, and adhesion, and reduces the number of LM8 lung metastases. This study provides evidence that reducing cell motility by increasing stiffness is a promising anti-metastatic strategy, shedding light on the development of novel osteosarcoma treatments. Overall, the manuscript is well-written and effectively addressed the study design, experimental methods, and findings. I would like to recommend to accept this manuscript after minor revisions. Several minor suggestions are listed as follows: 

1. All figures should be revised to better align with the style and format of a research article. This includes adjusting the figures to be more readable and visually appealing, as well as adding necessary labels and legends.

2. Please provide more detailed figure captions that describe the experimental details and results in more depth. This will help readers understand the findings presented in each figure.

3. Please provide a more comprehensive discussion of the risks of using CBX as a treatment for osteosarcoma, particularly in combination with other medications, as well as the potential drug-drug interactions associated with carbenoxolone (CBX) when used together with other therapeutic drugs. 

Comments on the Quality of English Language

The language is clear and concise, and the manuscript is well-organized, making it easy to follow the study design, methods, and results. There are no major issues with grammar or sentence structure. 

Author Response

Assigned Editor: Mr. Bruno Zhang

Journal: CIMB (ISSN 1467-3045)

Manuscript ID: cimb-2373991

Type: Article

Number of Pages: 16

Title: Carbenoxolone prevents lung metastasis of LM8 mouse osteosarcoma cells by regulating cell stiffness

Authors: Kouji Kita , Kunihiro Asanuma * , Takayuki Okamoto , Eiji Kawamoto , Koichi Nakamura , Tomohito Hagi , Tomoki Nakamura , Motomu Shimaoka , Akihiro Sudo

Abstract

Aim: Primary malignant bone tumor osteosarcoma occurs lung metastasis. Diminishing the lung metastasis, would have an enormous effect on the prognosis of patients. Previous our studies have demonstrated that the highly metastatic osteosarcoma cell lines had significantly softer than that of low metastatic cell line. Given this, we hypothesized that increase of cell stiffness might conduct metastasis suppression via reducing cell motility. In this study, we test whether carbenoxolone (CBX) increases LM8 osteosarcoma cell stiffness and prevents lung metastasis in vivo.

Methods: We evaluated actin cytoskeletal structure and polymerization of CBX-treated LM8 and by actin staining. Cell stiffness was measured using Atomic Force Microscopy (AFM). The metastasis related cell functions were analyzed by cell proliferation assay, wound healing assay, invasion assay, and cell adhesion assay. Furthermore, we administrated CBX into LM8-bearing mice and then evaluate the lung metastasis.

Results: Treatment with CBX significantly increased actin intensity and cell stiffness of LM8 compared with vehicle treatment LM8 (P<0.01). In young’s modulus images, compared to the control group, rigid fibrillate structures were observed in the CBX treatment group. CBX also suppressed cell migration, invasion, and adhesion, but not cell proliferation. Moreover, the number of LM8 lung metastasis were significantly reduced in the CBX administration group compared with the control group (P<0.01).

Conclusion: In this study, we demonstrate that CBX increases cell stiffness and significantly reduced lung metastasis. Our study provided first evidence that increase of cell stiffness leading to the reduction of cell motility might be effective as a novel anti-metastatic approach in vivo. 

Responses to the reviewers’ comments
Reviewer #3:

Comment 1:

All figures should be revised to better align with the style and format of a research article. This includes adjusting the figures to be more readable and visually appealing, as well as adding necessary labels and legends.

Response 1:

Thank you for your comment. We fixed the figures to be more readable.

We changed the font size and added the units on the Y axis in figure2a and 3a.

Comment 2:

Please provide more detailed figure captions that describe the experimental details and results in more depth. This will help readers understand the findings presented in each figure.

Response 2:

Thank you for your comment. We added the sentence.

Added details to each caption.

Comment 3:

Please provide a more comprehensive discussion of the risks of using CBX as a treatment for osteosarcoma, particularly in combination with other medications, as well as the potential drug-drug interactions associated with carbenoxolone (CBX) when used together with other therapeutic drugs. 

Response 3:

Thank you for your comment. In line 323-, we added the sentence about combination use of carbenoxolone.

Thank you for your comment.

We fixed manuscript again.

Reviewer 4 Report

Comments and Suggestions for Authors

The article entitled “Carbenoxolone Prevents lung metastasis of LM8 mouse osteosarcoma cells by regulating cell stiffness” describes the effect of Carbenoxolone in osteosarcoma and metastasis inhibition. The research work is fascinating and interesting. However, the author should address the following minor and significant concerns.

1) On Page 4 line 160 the authors described the use of C3H/He female mouse in the study. The authors need to justify using female animals in the study. Establish that there will be no sex hormone-dependent metabolic influence in C3H/He animal models.

MINOR corrections

1) Follow MDPI’s Instructions for Authors and updated journal template. This includes the title font format, the corresponding font style used in the manuscript” Palatino Linotype”, the author names font format, and the way the manuscript is structured. The authors have used the 2021 template use the updated template during the resubmission process. The figure legend should be followed by the corresponding figure. Keywords should be after the abstract.

2) There are some inconsistencies in font size on page 11; 247-253, font color inconsistencies 275-277, 278-282, 287-288 and 307-309.

3) Bar charts seem to be like an undergrad thesis rather than done by professionals; on page 6 figure 2a, page 7 figure 3a, mention the units on the Y axis. Keep the graph axis, labels, and numerals font uniform throughout the manuscript.

4) There are too many references this seems to be an article so limit the use of references to 25-30, Use appropriate references where necessary and remove the unnecessary ones.   

Comments on the Quality of English Language

1) The authors are suggested to change the title to grab the reader's attention. For E.g., “Carbenoxolone: A Novel Approach to Reducing Lung Metastasis in Osteosarcoma” or “A Novel Approach to Reducing Lung Metastasis in Osteosarcoma: Increasing Cell Stiffness with Carbenoxolone” or “Exploring the Therapeutic Potential of Carbenoxolone for Osteosarcoma Lung Metastasis” or “The Effect of Carbenoxolone on Actin Cytoskeletal Structure and Metastasis in Osteosarcoma”

Rephrase technical terms and avoid jargon doing so would interest the general audience. For E.g., on page 2 line 42: “Osteosarcoma is a common type of bone cancer. It is the most frequently occurring primary malignant tumor in bones.”

2) Uses shorter sentences and active voice to make the writing more engaging.

For E.g., Line 42 can be rewritten as: “Despite advances in treatment, half of all patients with osteosarcoma develop lung metastasis, which is a major cause of death.”

3) Use appropriate transition words or phrases where necessary to improve the reading flow. For E.g., Line 46 can be rewritten as: “Researchers have explored various approaches to prevent osteosarcoma metastasis, such as targeting specific molecules. However, the prognosis for patients with metastatic osteosarcoma has not improved in 30 years.”

Author Response

Assigned Editor: Mr. Bruno Zhang

Journal: CIMB (ISSN 1467-3045)

Manuscript ID: cimb-2373991

Type: Article

Number of Pages: 16

Title: Carbenoxolone prevents lung metastasis of LM8 mouse osteosarcoma cells by regulating cell stiffness

Authors: Kouji Kita , Kunihiro Asanuma * , Takayuki Okamoto , Eiji Kawamoto , Koichi Nakamura , Tomohito Hagi , Tomoki Nakamura , Motomu Shimaoka , Akihiro Sudo

Abstract

Aim: Primary malignant bone tumor osteosarcoma occurs lung metastasis. Diminishing the lung metastasis, would have an enormous effect on the prognosis of patients. Previous our studies have demonstrated that the highly metastatic osteosarcoma cell lines had significantly softer than that of low metastatic cell line. Given this, we hypothesized that increase of cell stiffness might conduct metastasis suppression via reducing cell motility. In this study, we test whether carbenoxolone (CBX) increases LM8 osteosarcoma cell stiffness and prevents lung metastasis in vivo.

Methods: We evaluated actin cytoskeletal structure and polymerization of CBX-treated LM8 and by actin staining. Cell stiffness was measured using Atomic Force Microscopy (AFM). The metastasis related cell functions were analyzed by cell proliferation assay, wound healing assay, invasion assay, and cell adhesion assay. Furthermore, we administrated CBX into LM8-bearing mice and then evaluate the lung metastasis.

Results: Treatment with CBX significantly increased actin intensity and cell stiffness of LM8 compared with vehicle treatment LM8 (P<0.01). In young’s modulus images, compared to the control group, rigid fibrillate structures were observed in the CBX treatment group. CBX also suppressed cell migration, invasion, and adhesion, but not cell proliferation. Moreover, the number of LM8 lung metastasis were significantly reduced in the CBX administration group compared with the control group (P<0.01).

Conclusion: In this study, we demonstrate that CBX increases cell stiffness and significantly reduced lung metastasis. Our study provided first evidence that increase of cell stiffness leading to the reduction of cell motility might be effective as a novel anti-metastatic approach in vivo. 

Responses to the reviewers’ comments
Reviewer #4:

Comment 1:

The article entitled “Carbenoxolone Prevents lung metastasis of LM8 mouse osteosarcoma cells by regulating cell stiffness” describes the effect of Carbenoxolone in osteosarcoma and metastasis inhibition. The research work is fascinating and interesting. However, the author should address the following minor and significant concerns.

1) On Page 4 line 160 the authors described the use of C3H/He female mouse in the study. The authors need to justify using female animals in the study. Establish that there will be no sex hormone-dependent metabolic influence in C3H/He animal models.

Response 1:

Thank you for your comment.

Osteosarcoma usually develops in adolescent. Sex hormones have a potential to be related to disease progression. Sex steroid receptors including estrogen and androgen were detected in osteosarcoma tissues and cell lines (Dohi. Cancer science. 2008. https://doi.org/10.1111/j.1349-7006.2007.00673.x). There many reports about osteosarcoma and sex hormones including androgen and estrogen. By a big data analysis of 2849 high grade osteosarcoma patients from SEER, 10 year survival in male was significantly worse (54.5%) than female (60.9%) (p<0.05) (Duchman. Cancer Epidemiology. 2015. https://doi.org/10.1016/j.canep.2015.05.001). This big data may leads to the result that sex hormone in male had worse effect than in female. However, the effect of sex hormones on osteosarcoma patients is controversial.

In this time, we used female mice because male mice are aggressive and sometimes tumors on the back were lost by fighting and size calculation was not performed.

Resveratrol Resveratrol Down-Regulates the Androgen Receptor

Harada et al. J Nutr Sci Vitaminol

. 2007 Dec;53(6):556-60. doi: 10.3177/jnsv.53.556.

Finally, oral administration of RSV (25 mg/kg, twice daily) for 30 days to mice injected with the highly metastatic osteosarcoma LM8 cells significantly reduced the number of metastatic tumors in both the lungs and the liver. Kimura et al, Nutr Cancer. 2016 May-Jun;68(4):667-78. doi: 10.1080/01635581.2016.1158295.

Comment 2:

Follow MDPI’s Instructions for Authors and updated journal template. This includes the title font format, the corresponding font style used in the manuscript” Palatino Linotype”, the author names font format, and the way the manuscript is structured. The authors have used the 2021 template use the updated template during the resubmission process. The figure legend should be followed by the corresponding figure. Keywords should be after the abstract.

Response 2:

Thank you for your comment. We followed journal template. Keywords moved after the abstract.

Comment 3:

There are some inconsistencies in font size on page 11; 247-253, font color inconsistencies 275-277, 278-282, 287-288 and 307-309.

Response 3:

Thank you for your comment. We changed the font size and font color.

Comment 4:

Bar charts seem to be like an undergrad thesis rather than done by professionals; on page 6 figure 2a, page 7 figure 3a, mention the units on the Y axis. Keep the graph axis, labels, and numerals font uniform throughout the manuscript.

Response 4:

Thank you for your comment. Added the units on the Y axis in figure 2 and figure 3.

Comment 5:

There are too many references this seems to be an article so limit the use of references to 25-30, Use appropriate references where necessary and remove the unnecessary ones.   

Response 5:

Thank you for your comment. We removed the unnecessary references.

Comment 6:

The authors are suggested to change the title to grab the reader's attention. For E.g., “Carbenoxolone: A Novel Approach to Reducing Lung Metastasis in Osteosarcoma” or “A Novel Approach to Reducing Lung Metastasis in Osteosarcoma: Increasing Cell Stiffness with Carbenoxolone” or “Exploring the Therapeutic Potential of Carbenoxolone for Osteosarcoma Lung Metastasis” or “The Effect of Carbenoxolone on Actin Cytoskeletal Structure and Metastasis in Osteosarcoma”

Rephrase technical terms and avoid jargon doing so would interest the general audience. For E.g., on page 2 line 42: “Osteosarcoma is a common type of bone cancer. It is the most frequently occurring primary malignant tumor in bones.”

Response 6:

Thank you for your comment. We changed the title.

“A Novel Approach to Reducing Lung Metastasis in Osteosarcoma: Increasing Cell Stiffness with Carbenoxolone”

Comment 7:

Uses shorter sentences and active voice to make the writing more engaging.

For E.g., Line 42 can be rewritten as: “Despite advances in treatment, half of all patients with osteosarcoma develop lung metastasis, which is a major cause of death.”

Response 7:

Thank you for your comment. We fixed as pointed out.

Comment 8:

Use appropriate transition words or phrases where necessary to improve the reading flow. For E.g., Line 46 can be rewritten as: “Researchers have explored various approaches to prevent osteosarcoma metastasis, such as targeting specific molecules. However, the prognosis for patients with metastatic osteosarcoma has not improved in 30 years.”

Response 8:

Thank you for your comment. We fixed as pointed out.

Thank you for your comment.

We fixed manuscript again.

Round 2

Reviewer 1 Report

Comments and Suggestions for Authors

The authors carefully addressed the points raised by the previous reviewers and, from my side, I don't have any additional observation. The manuscript, in my opinion, is ready for publication.